# Periostin Contributes to Fibrocartilage Layer Growth of the Patella Tendon Tibial Insertion in Mice

**DOI:** 10.3390/medicina58070957

**Published:** 2022-07-19

**Authors:** Hirotaka Mutsuzaki, Yuta Yoshida, Hiromi Nakajima

**Affiliations:** 1Department of Orthopedic Surgery, Ibaraki Prefectural University of Health Sciences, 4669-2 Ami Ami-machi, Inashiki-gun, Ibaraki, Ami 300-0394, Japan; 2Department of Agriculture, Ibaraki University, 3-21-1 Chuo, Ibaraki, Ami 300-0393, Japan; yuta.yoshida.animal@vc.ibaraki.ac.jp (Y.Y.); hiromi.nakajima.vmd@vc.ibaraki.ac.jp (H.N.)

**Keywords:** patella tendon insertion, fibrocartilage layers, periostin, growth, knockout mouse

## Abstract

*Background and Objectives:* The influence of periostin on the growth of the patella tendon (PT) tibial insertion is unknown. The research described here aimed to reveal the contribution of periostin to the growth of fibrocartilage layers of the PT tibial insertion using periostin knockout mice. *Materials and Methods:* In both the wild-type (WD; C57BL/6N, periostin +/+; *n* = 54) and periostin knockout (KO; periostin −/−; *n* = 54) groups, six mice were euthanized on day 1 and at 1, 2, 3, 4, 6, 8, 10, and 12 weeks of age. Chondrocyte proliferation and apoptosis, number of chondrocytes, safranin O-stained glycosaminoglycan (GAG) area, staining area of type II collagen, and length of the tidemark were investigated. *Results:* Chondrocyte proliferation and apoptosis in KO were lower than those in WD on day 1 and at 1, 4, and 8 weeks and on day 1 and at 4, 6, and 12 weeks, respectively. Although the number of chondrocytes in both groups gradually decreased, it was lower in KO than in WD on day 1 and at 8 and 12 weeks. In the extracellular matrix, the GAG-stained area in KO was smaller than that in WD on day 1 and at 1, 4, 8, 10, and 12 weeks. The staining area of type II collagen in KO was smaller than that in WD at 8 weeks. The length of the tidemark in KO was shorter than that in WD at 4 and 6 weeks. *Conclusion:* Loss of periostin led to decreased chondrocyte proliferation, chondrocyte apoptosis, and the number of chondrocytes in the growth process of the PT tibial insertion. Moreover, periostin decreased and delayed GAG and type II collagen production and delayed tidemark formation in the growth process of the PT tibial insertion. Periostin can, therefore, contribute to the growth of fibrocartilage layers in the PT tibial insertion. Periostin deficiency may result in incomplete growth of the PT tibial insertion.

## 1. Introduction

Periostin, a secreted profibrogenic glycoprotein, is a matricellular protein that has a regulatory role in disease processes, healing, and development [1]. Periostin is located in the extracellular matrix (ECM) and regulates cell migration and proliferation [2]. Periostin is localized in the ECM of the periosteum and periodontal ligament [3], as well as in heart valves, tendons, wounds, and tumors, and is re-expressed after skeletal muscle, myocardial, and vascular injuries, as well as after bone fractures [1,4,5]. Periostin plays a role in proliferation, apoptosis, ECM synthesis, collagen fibrillogenesis, growth factor production, and cell morphology [4]. Periostin ensures tissue strength by controlling collagen fiber formation through the regulation of type I collagen’s cross-linked structures and ECM uptake of tenascin C [6,7,8,9,10].

Tendon and ligament insertions have fibrocartilage layers between the soft and hard tissue, as is the case in the anterior cruciate ligament (ACL) and the patella tendon (PT) tibial insertions [11]. The ECM of fibrocartilage layers contains type II collagen and glycosaminoglycans (GAGs) that enables it to resist tensile, shear, and compressive stresses, thereby transmitting load [11,12]. Periostin has been shown to influence the growth of the fibrocartilage layer of the ACL insertion in periostin knockout mice [13]. Periostin also decreased cell proliferation in the early growth phase and delayed the development of the ECM in the fibrocartilage layer of the ACL insertion in periostin knockout mice [13].

Osgood–Schlatter disease is one of the PT tibial insertional tendinopathies that is clinically well-known for the prolonged treatment it requires [14]. Although the PT tibial insertion is similar to the formation of the ACL insertion, the mechanical environment is different in that one side is fixed to the bone and the other side to the muscle via the patella. Therefore, periostin may function differently in the PT tibial insertion compared to the ACL insertion. The influence of periostin on PT tibial insertion growth is unknown.

Therefore, the purpose of this study was to assess the contribution of periostin to the growth of fibrocartilage layers of the PT tibial insertion using periostin knockout mice. We hypothesized that periostin would influence the growth of fibrocartilage layers of the PT tibial insertion. The influence of periostin on the growth of the patellar tendon insertion is expected to be important in the repair and regeneration of periostin on the patellar tendon insertion.

## 2. Materials and Methods

### 2.1. Animal Preparation

Since this research could not be undertaken in vitro and since mice are the smallest experimental animals, mice were selected for this study. Periostin knockout mice were created as described in our previous report [13]. Four C57BL/6N periostin knockout (periostin +/−) mice (female: 1, male: 3) were created by Cyagen Biosciences Inc., Suzhou, Jiangsu, China. To create periostin knockout (periostin −/−) mice (KO), four periostin knockout (periostin +/−) mice and eight C57BL/6N wild-type (periostin +/+) mice (WD; female: 8, Japan SLC, Inc., Hamamatsu, Shizuoka, Japan) were allowed to mate, and 54 KO mice were obtained [13]. To create WD, six C57BL/6N WD (female: 3, male: 3, Japan SLC, Inc., Hamamatsu, Shizuoka, Japan) were allowed to mate, and 54 WD mice were obtained [13]. Groups of five mice each were bred in a polycarbonate cage with paper bedding material at 25 °C under 12 h light/dark cycles. The mice ate feed and drank water and could move freely in their cages. No adverse events occurred. At 3 months, the skeletal growth of the mice is complete [15], therefore, the final investigation period was set as 12 weeks of age. On d 1 and at 1, 2, 3, 4, 6, 8, 10, and 12 weeks of age, six mice in each group were euthanized by cervical dislocation.

The genetic confirmation of periostin knockout by polymerase chain reaction (PCR) analysis of isolated DNA was performed using tail DNA according to our previous report [13]. NucleoSpin^®^ Tissue (Takara Bio Inc., Kusatsu, Shiga, Japan) and TaKaRa TaqTM Hot Start Version (Takara Bio Inc., Kusatsu, Shiga, Japan) were used. Forward primer was (F1):5′-TGAAGCTACCCATCTCCCAAATG-3′ and reverse primers were (R1):5′-CCTCTCCCAGCGTTCATAAATC-3′ and (R2):5′-ACCATCCTGTAGGCTCTTCAAAC-3 (Eurofins Genomics, Tokyo, Japan). Under the conditions described in the Mouse Conventional Knockout User Manual (Cyagen Biosciences Inc., Silicon Valley, CA, USA), PCR was performed using a thermal cycler (MiniAmp Plus, Thermo Fisher Scientific, Waltham, MA, USA) [13].

The mice were kept in accordance with the guidelines of the institution’s Ethical Committee and the National Institutes of Health (NIH) Guidelines for the Care and Use of Laboratory Animals (NIH pub. No. 86–23, rev. 1985). This study conformed to the Animal Research: Reporting In Vivo (ARRIVE) guidelines.

### 2.2. Staining Method and Immunohistochemistry

Specimens were fixed with 10% neutral-buffered formalin, decalcified, and embedded in paraffin in accordance with our previous report [13]. All the specimens were sliced 5 µm in the sagittal plane at unilateral knees [13].

Hematoxylin and eosin and safranin O staining were performed to evaluate the histomorphology and GAG production [13]. To distinguish proliferating cells, proliferating cell nuclear antigen (PCNA) staining using a Histofine^®^ SAB-PO (M) Kit (Nichirei Biosciences Inc., Tokyo, Japan), an anti-PCNA monoclonal antibody (PC-10; code No. M0879; Dako, Glostrup, Denmark), and an antibody diluent (code No. S0809; Dako) was performed (Figure 1A) [13]. To distinguish apoptotic cells, terminal deoxynucleotidyl transferase-mediated deoxyuridine triphosphate-biotin nick-end labeling (TUNEL) staining using an Apoptag^®^ Plus Peroxidase In Situ Apoptosis Detection Kit (Merck Millipore, Billerica, MA, USA) was performed (Figure 1B) [13]. To distinguish type II collagen, a Histofine^®^ SAB-PO (M) Kit (Nichirei Biosciences Inc., Tokyo, Japan) with an anti-type II collagen monoclonal antibody (Kyowa Pharma Chemical, Toyama, Japan) was used (Figure 1C) [13].

### 2.3. Histomorphometric Analysis

Histomorphometric analysis was performed in accordance with our previous report [13]. Lower-density stained cartilaginous tissues with round cells between the hyaline cartilage area and the ligament were identified as fibrocartilage layers on d 1 and at 1 and 2 weeks of age [13]. In the other specimens, fibrocartilage layers were identified as layers with round cells between the ligament and the bone [13]. The border between the tendon and the fibrocartilage layers was identified by spindle-shaped cells and round cells [13]. In the fibrocartilage layers of the PT tibial insertion, regions stained red by safranin O were identified as GAG production areas (Figure 1D) [13]. Regions stained brown were identified as type II collagen [13]. The total tidemark length was measured using hematoxylin and eosin staining (Figure 1E) [13]. The width of the PT tibial insertion at the level of the fibrocartilage layers was measured. To observe and measure the chondrocytes, GAG production areas, type II collagen-stained areas, and tidemark length in the PT tibial insertion, a BX-51 light microscope (Olympus Optical Co. Ltd., Tokyo, Japan) and Mac Scope software (Mitani Co., Fukui, Japan) were used. The total number of chondrocytes, red-stained GAG production areas, type II collagen-stained areas, and tidemark lengths were divided by the width of the PT tibial insertion. Subsequently, the number of chondrocytes per width of insertion, thickness of GAG production areas, thickness of type II collagen-stained areas, and percentage of tidemark length were calculated [13]. The percentages of TUNEL- and PCNA-positive chondrocytes and the number of positive chondrocytes were calculated [13]. At different ages, those parameters were compared to the parameters obtained at the age of 12 weeks and between the WD and KO groups [13].

### 2.4. Statistical Analysis

Since two-way analysis of variance (ANOVA) revealed that the interactions were significant, we decided to compare each parameter individually after the Shapiro–Wilk normality test was performed for each parameter. When all the variables for each parameter were normally distributed, Student’s *t*-test was performed to compare WD with KO. When all the variables for each parameter were not normally distributed, the Mann–Whitney *U* test was performed to compare the data of WD and KO. To evaluate the time-dependent changes compared with those at 12 weeks of age, one-way ANOVA and Dunnett’s test were used. *p*-values of less than 0.05 were considered statistically significant. The statistical analyses were performed using IBM SPSS Statistics version 28.0 (IBM Corp., Armonk, NY, USA).

Using the POWER procedure in SAS software (SAS Institute, Cary, NC, USA), power analysis was conducted with a confidence level of 95% (α = 0.05) and power (1–β) of 80% with reference to previous research [16,17]. The smallest sample size was calculated at 5–6 specimens per age group. Therefore, six specimens per age group were enrolled.

## 3. Results

The summary of histomorphometric analyses is shown in Table 1.

The chondrocyte proliferation rate (Figure 2) of KO was lower than that of WD on d 1 and at 1, 4, and 8 weeks. In WD, the chondrocyte proliferation rate was lower on d 1 than at 12 weeks, and in KO, the rate was lower on d 1 and at 1, 4, and 8 weeks than at 12 weeks.

The chondrocyte apoptosis rate (Figure 3) in KO was lower than that in WD on d 1 and at 4, 6, and 12 weeks. In WD, the chondrocyte apoptosis rate was higher at 1, 4, 6, and 8 weeks than at 12 weeks, and in KO, the rate was higher at 1, 2, 3, 4, 6, and 8 weeks than at 12 weeks.

The number of chondrocytes per width of insertion (Figure 4) in KO was lower than that in WD on d 1 and at 8 and 12 weeks but higher at 2 weeks. In WD, the number of chondrocytes per width of insertion was higher on d 1 and at 1 week than at 12 weeks, and in KO, the rate was higher on 1 d 1 and at 1 and 2 weeks than at 12 weeks.

The thickness of safranin O-stained GAG areas (Figure 5) in KO was lesser than that in WD on d 1 and at 1, 4, 8, 10, and 12 weeks but greater at 2 weeks. In WD, the thickness of safranin O-stained GAG areas was greater on d 1 and at 1 week than at 12 weeks, and in KO, the number was greater at 1, 2, and 6 weeks than at 12 weeks.

The thickness of type II collagen staining areas (Figure 6) in KO was lesser than that in WD at 8 weeks. In WD, the thickness of type II collagen staining areas was lesser on d 1 and at 1, 2, 3, and 4 weeks than at 12 weeks, and in KO, the thickness was lesser on d 1 and at 1, 2, and 3 weeks than at 12 weeks.

The percentage of the tidemark length (Figure 7) in KO was lower than that in WD at 4 and 6 weeks but higher at 12 weeks. In WD, the percentage of the tidemark length was lower on d 1 and at 1, 2, and 3 weeks than at 12 weeks, and in KO, the percentage was lower on d 1 and at 1, 2, 3, and 4 weeks than at 12 weeks.

The width of insertion (Figure 8) in KO was lesser than that in WD at 1 and 4 weeks. In WD, the width of insertion was lesser on d 1 than at 12 weeks, and in KO, the width was greater on d 1 and at 1, 2, and 4 weeks than at 12 weeks.

## 4. Discussion

Although the number of chondrocytes in both groups gradually decreased, chondrocyte proliferation, chondrocyte apoptosis, and the number of chondrocytes were low from early birth to the end of growth of the PT tibial insertion in KO. Moreover, the GAG production and type II collagen areas and the tidemark length in KO were smaller than those in WD at 12 weeks, 8 weeks, and 4 and 6 weeks, respectively.

The GAG-stained area was large from day 1 to week 1 in WD and from week 1 to week 6 in KO. The GAG-stained area in KO was lesser than that in WD on day 1 and at weeks 1, 4, 8, 10, and 12. Absence of periostin can delay and decrease GAG production. The type II collagen staining area in KO was lesser than that in WD at 8 weeks. Absence of periostin can also decrease type II collagen production. Although chondrocyte proliferation and increased GAG-stained areas were observed in the early stage of growth, it has been reported that type II collagen is observed after calcified and uncalcified cartilage layers are formed at the insertion [12]. Therefore, type II collagen was expected to have been observed in the second half of the growth process. The percentage of the tidemark length in KO was lower than that in WD at 4 and 6 weeks. Absence of periostin can delay tidemark production. Periostin plays an important role in the growth of the ECM at the PT tibial insertion and prevents complete growth.

Chondrocyte proliferation in KO was lower than that in WD on day 1 and at 1, 4, and 8 weeks. Chondrocyte apoptosis in KO was also lower than that in WD on day 1 and at 4, 6, and 12 weeks. Although the number of chondrocytes gradually decreased in both groups, the number of chondrocytes in KO was also lower than that in WD on day 1 and at 2, 8, and 12 weeks. Periostin may upregulate chondrocyte proliferation and apoptosis in the PT tibial insertion. The imbalance between chondrocyte proliferation and apoptosis can affect the number of chondrocytes. As a result, it can lead to delayed and incomplete ECM growth at the PT tibial insertion. In PT tibial insertion growth, periostin is thought to be an important regulator.

In comparison with a previous study that reported ACL insertion growth using periostin KO mice [13], significant differences were observed in chondrocyte proliferation, chondrocyte apoptosis, and the number of chondrocytes, even from 4 weeks to 12 weeks in this study. Moreover, tidemark length of the PT insertion showed a significant difference earlier than that of the ACL insertion, and GAG production showed a significant difference at 8, 10, and 12 weeks in this study. Other phenomena were similar in both the ACL and PT insertions. Both ACL and PT tibial insertions are direct-type insertions that include four transitional tissue layers, ligament or tendon, two fibrocartilage layers (uncalcified and calcified), and bone, that transmit mechanical stress [11]. However, in the ACL, both ends are made up of bone, whereas one end of the PT tibial insertion is a muscle through the patella. The differences between the ACL and PT tibial insertions can be due to the differences in structure and mechanical environments. It has been reported that the actual 1/3 width of the bone–PT–bone complex has a higher ultimate load than the femur–ACL–tibia complex [18,19]. Therefore, the PT tibial insertion is affected by muscle traction in the growth period and may be subjected to greater tensile stresses than the ACL insertion. Moreover, periostin is sensitive to mechanical stress and controls bone modeling and remodeling [20]. Periostin is involved in the growth of the ACL insertion and the PT tibial insertion in a mechanical environment; however, periostin may have a different influence on the ACL insertion than the PT tibial insertion where the mechanical load is larger.

In terms of clinical diseases, Osgood–Schlatter disease is a well-known tendinopathy involving the PT tibial insertion [14], which requires prolonged treatment. Periostin has been reported to be upregulated by growth factors such as BMP-2, basic FGF, TGF-β, and platelet-derived growth factor [5]. Periostin is upregulated during muscle regeneration [21] and fracture healing [22] and is necessary to regenerate tendons [23]. The previously mentioned growth factors enhanced tendon-to-bone healing [24]. Moreover, dynamic tensile stimulation may be an essential factor for tendon regeneration [25]. In bone modeling and remodeling, mechanical stress plays an important role, and periostin is expected to be involved in controlling these systems in bone [20]. Therefore, it is possible that periostin influences growth factors and is involved in the growth, maintenance, and regeneration of tendons and bones in a mechanical environment. Periostin may be a candidate to treat tendinopathies such as Osgood–Schlatter disease. Moreover, periostin may contribute as a marker of the repair/healing process and a predictor of treatment response and tendon-to-bone healing.

The limitations of this study need to be noted. Although the complete skeletal growth of mice is achieved at 3 months [15], evaluations after 3 months of age might be necessary to investigate the effects after growth. Moreover, mechanical evaluations are necessary to elucidate the influence of mechanical loads.

## 5. Conclusions

Periostin decreased chondrocyte proliferation, chondrocyte apoptosis, and the number of chondrocytes in the growth process of the PT tibial insertion. Moreover, periostin decreased and delayed GAG and type II collagen production and tidemark formation in the growth process of the PT tibial insertion. Periostin can contribute to the growth of fibrocartilage layers in the PT tibial insertion. Periostin deficiency may result in incomplete PT tibial insertion growth.

## Figures and Tables

**Figure 1 medicina-58-00957-f001:**
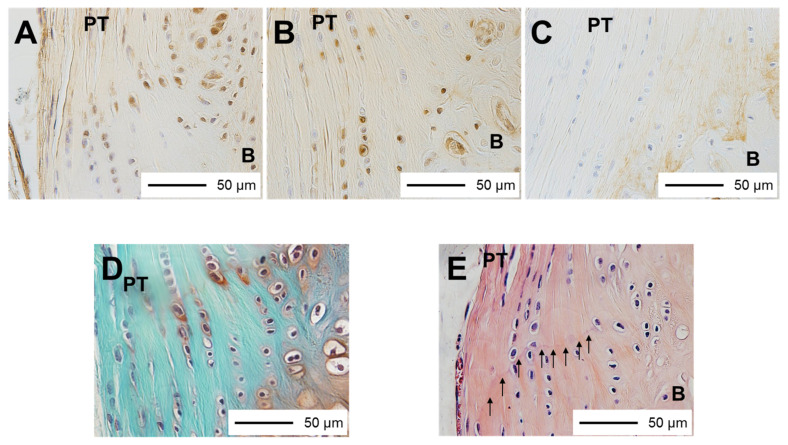
Tissue specimens: (**A**) PCNA staining, PCNA-positive cells were brown (400×); (**B**) TUNEL staining, TUNEL-positive cells were brown (400×); (**C**) type II collagen staining, brown area is type II collagen-stained area (400×); (**D**) safranin O staining, red area is glycosaminoglycan production area (400×); (**E**) HE staining, the tidemark is between the calcified and uncalcified fibrocartilage layers (arrows) (400×). PT, patella tendon; B, bone; PCNA, proliferating cell nuclear antigen; TUNEL, terminal deoxynucleotidyl transferase-mediated deoxyuridine triphosphate-biotin nick-end labeling.

**Figure 2 medicina-58-00957-f002:**
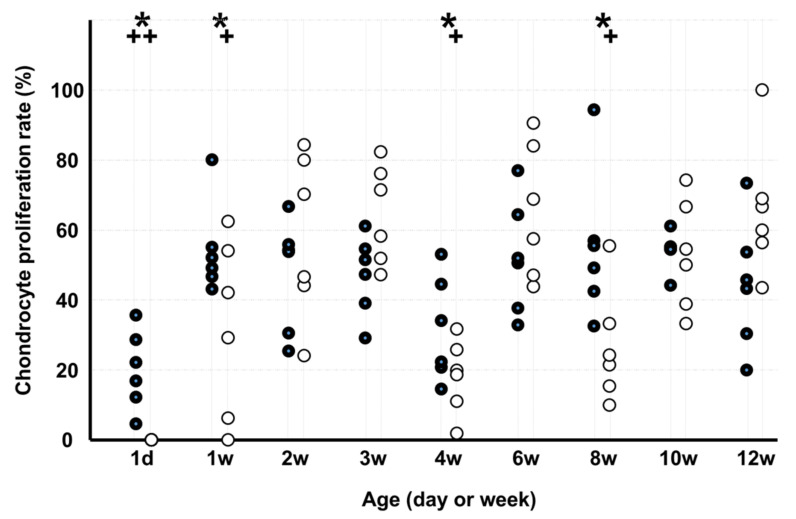
Chondrocyte proliferation rate. Note: * *p* < 0.05, WD vs. KO (*n* = 6); + *p* < 0.05, vs. age of 12 weeks (*n* = 6); ●: WD; ◯: KO. Some markers may be overlapping.

**Figure 3 medicina-58-00957-f003:**
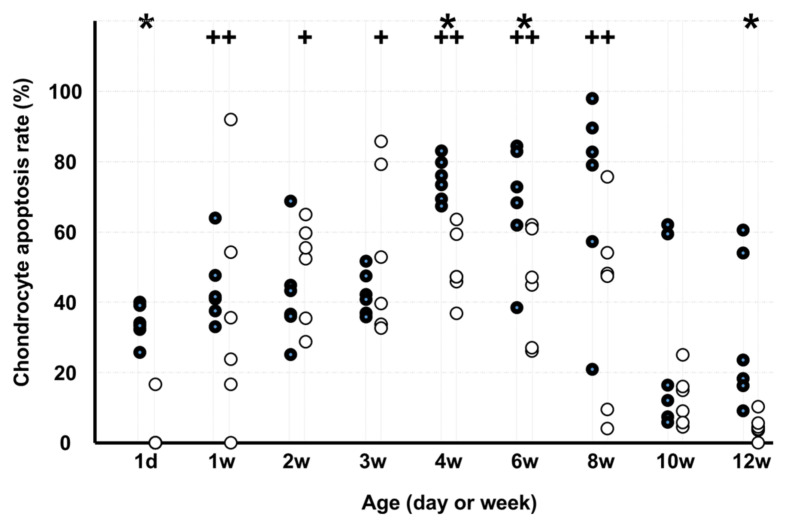
Chondrocyte apoptosis rate. Note: * *p* < 0.05, WD vs. KO (*n* = 6); + *p* < 0.05, vs. age of 12 weeks (*n* = 6); ●: WD; ◯: KO. Some markers may be overlapping.

**Figure 4 medicina-58-00957-f004:**
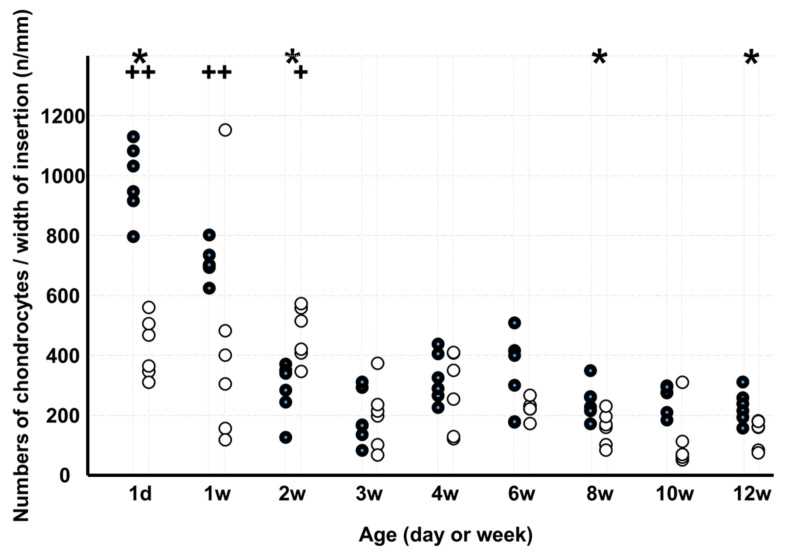
Numbers of chondrocytes per width of insertion. Note: * *p* < 0.05, WD vs. KO (*n* = 6); + *p* < 0.05, vs. age of 12 weeks (*n* = 6); ●: WD; ◯: KO. Some markers may be overlapping.

**Figure 5 medicina-58-00957-f005:**
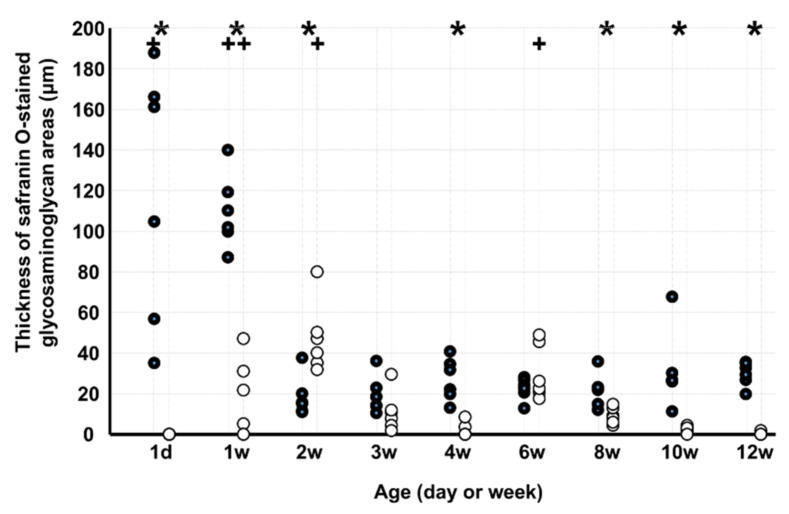
Thickness of safranin O-stained glycosaminoglycan areas. Note: * *p* < 0.05, WD vs. KO (*n* = 6); + *p* < 0.05, vs. age of 12 weeks (*n* = 6); ●: WD; ◯: KO. Some markers may be overlapping.

**Figure 6 medicina-58-00957-f006:**
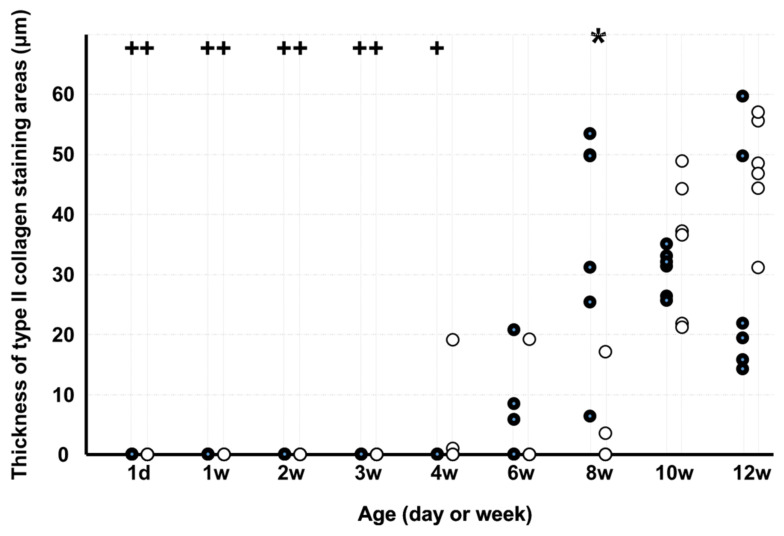
Thickness of type II collagen staining areas. Note: * *p* < 0.05, WD vs. KO (*n* = 6); + *p* < 0.05, vs. age of 12 weeks (*n* = 6); ●: WD; ◯: KO. Some markers may be overlapping.

**Figure 7 medicina-58-00957-f007:**
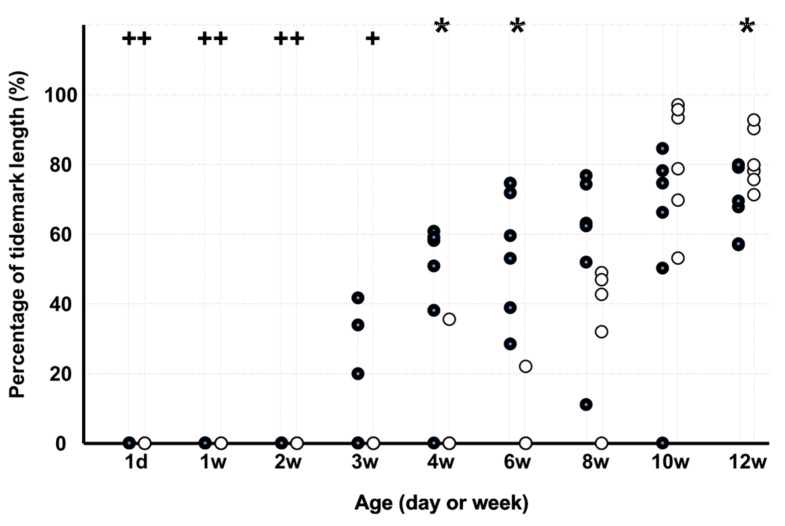
Percentage of the tidemark length. Note: * *p* < 0.05, WD vs. KO (*n* = 6); + *p* < 0.05, vs. age of 12 weeks (*n* = 6); ●: WD; ◯: KO. Some markers may be overlapping.

**Figure 8 medicina-58-00957-f008:**
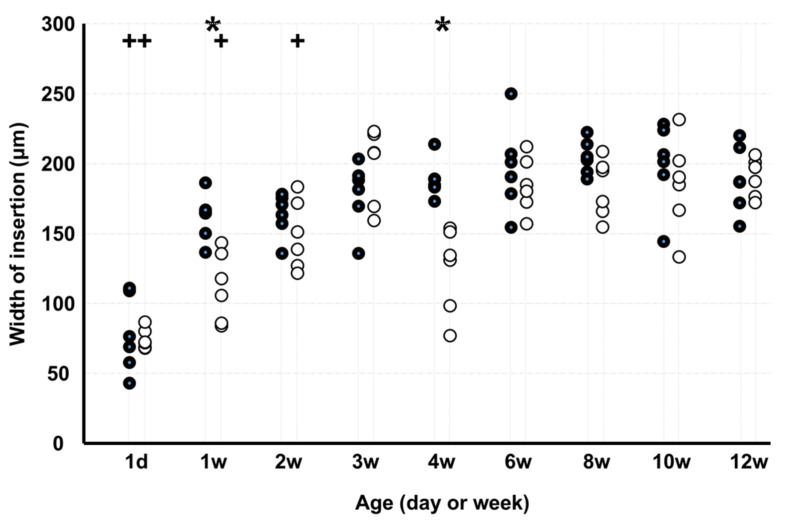
Width of insertion. Note: * *p* < 0.05, WD vs. KO (*n* = 6); + *p* < 0.05, vs. age of 12 weeks (*n* = 6); ●: WD; ◯: KO. Some markers may be overlapping.

**Table 1 medicina-58-00957-t001:** Summary of histomorphometric analyses.

		1d (*n* = 6)	1w (*n* = 6)	2w (*n* = 6)	3w (*n* = 6)	4w (*n* = 6)	6w (*n* = 6)	8w (*n* = 6)	10w (*n* = 6)	12w (*n* = 6)	*η* ^2^
Chondrocyte proliferation rate (%)	WD	20.0 ± 11.2 †	50.6 (47.3–54.3)	47.7 ± 16.1	47.1 ± 11.5	31.5 ± 15.0	52.4 ± 16.4	55.1 ± 21.2	54.1 ± 5.5	44.4 ± 18.5	0.406
	KO	0 †	32.4 ± 25.4 †	58.3 ± 23.6	64.6 ± 14.1	18.2 ± 10.6 †	65.3 ± 19.2	26.7 ± 16.2 †	53.0 ± 15.7	65.9 ± 19.0	0.670
	*p* value	0.002 *	0.012 *	n.s.	n.s.	<0.001 *	n.s.	0.002 *	n.s.	n.s.	
Chondrocyte apoptosis rate (%)	WD	34.1 ± 5.2	44.0 ± 10.8 †	42.4 ± 14.6	42.4 ± 6.1	74.8 ± 6.0 †	68.1 ±16.9 †	71.1 ± 28.2 †	14.2 (8.6–48.6)	30.3 ± 21.5	0.547
	KO	0.0 (0.0–4.2)	37.1 ± 32.5 †	49.5 ± 14.2 †	54.0 ± 23.3 †	50.0 ± 9.8 †	44.7 ± 15.7 †	39.9 ± 27.6 †	12.6 ± 7.7	4.7 ± 3.4	0.569
	*p* value	0.002 *	n.s.	n.s.	n.s.	<0.001 *	0.032 *	n.s.	n.s.	0.033 *	
Numbers of chondrocytes/width of insertion (n/mm)	WD	983.1 ± 122.3 †	708.4 ± 57.9 †	285.8 ± 91.5	192.4 ± 90.0	324.2 ± 82.2	329.4 ± 134.7	247.3 ± 59.8	256.1 ± 47.5	228.1 ± 53.5	0.910
	KO	426.4 ± 99.3 †	436.2 ± 337.7 †	470.6 ± 91.5 †	198.8 ± 108.1	279.4 ± 131.5	222.8 ± 30.4	157.3 ± 55.6	69.8 (62.7–102.2)	140.7 ± 47.7	0.473
	*p* value	<0.001 *	n.s.	0.006 *	n.s.	n.s.	n.s.	0.022 *	n.s.	0.014 *	
Thickness of safranin O–stained glycosaminoglycan areas (μm)	WD	118.5 ± 62.9 †	109.6 ± 18.3 †	15.4 (12.4–19.0)	19.3 ± 9.2	27.0 ± 10.4	22.6 ± 5.5	21.8 ± 8.2	26.5 (26.0–29.2)	29.9 ± 6.0	0.750
	KO	0	17.6 ± 19.3 †	47.5 ± 17.4 †	11.2 ± 9.8	0.0 (0.0–2.9)	30.4 ± 13.5†	9.3 ± 3.9	2.4 ± 1.6	0.7 ± 0.8	0.718
	*p* value	0.002 *	<0.001 *	0.009 *	n.s.	0.002 *	n.s.	0.007 *	0.002 *	<0.001 *	
Thickness of type II collagen staining areas (μm)	WD	0 †	0 †	0 †	0†	0†	5.8 ± 8.1	36.0 ± 18.5	30.6 ± 3.8	30.2 ± 19.5	0.751
	KO	0 †	0 †	0 †	0†	0.0 (0.0–0.8)	0.0 (0.0–0.0)	0.0 (0.0–2.7)	35.1 ± 11.4	47.3 ± 9.3	0.888
	*p* value	n.s.	n.s.	n.s.	n.s.	n.s.	n.s.	0.004 *	n.s.	n.s.	
Percentage of tidemark length (%)	WD	0 †	0 †	0 †	15.9 ± 18.8	54.5 (41.2–58.8)	54.4 ± 18.2	56.6 ± 24.0	58.9 ± 31.2	68.4 ± 10.0	0.734
	KO	0 †	0 †	0 †	0†	0.0 (0.0–0.0)	0.0 (0.0–0.0)	28.4 ± 22.8	81.3 ± 17.5	81.3 ± 8.4	0.906
	*p* value	n.s.	n.s.	n.s.	n.s.	0.015 *	0.002 *	n.s.	n.s.	0.036 *	
Width of insertion (μm)	WD	77.5 ± 27.4 †	161.4 ± 16.8	163.3 ± 15.5	178.1 ± 23.7	188.6 ± 13.5	196.7 ± 31.9	204.3 ± 12.3	199.3 ± 30.3	188.5 ± 24.1	0.756
	KO	74.7 ± 7.2 †	112.2 ± 24.8 †	148.9 ± 24.6 †	198.1 ± 27.1	124.3 ± 30.6 †	184.7 ± 19.8	182.4 ± 21.2	184.8 ± 33.0	190.0 ± 13.7	0.779
	*p* value	n.s.	0.002 *	n.s.	n.s.	<0.001 *	n.s.	n.s.	n.s.	n.s.	

Results are presented as the mean ± SD, if normality was confirmed. Results are presented as the median (interquartile range), if normality was not confirmed. * *p* < 0.05: significant difference between WD and KO. † *p* < 0.05 compared with the value of 12 weeks; KO, periostin knockout mice; WD, wild type mice.

## Data Availability

The datasets used and/or analyzed during the current study are available from the corresponding author on reasonable request.

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
