# Peer review of "Periostin Contributes to Fibrocartilage Layer Growth of the Patella Tendon Tibial Insertion in Mice"

_medicina, 2022, doi:10.3390/medicina58070957_

Round 1

Reviewer 1 Report

- The approach is interesting and the topic is appropriate for the journal.

-        The work  has a very clear structure and all the sections are well written in a way that is easy to read and understand.

-         However, little modifications and improvements are needed to enhance the quality of the paper.

-        The paper deals with the Contribution of Periostin to Fibrocartilage Layer Growth of Patella Tendon Tibial Insertion in Mice, reporting interesting results. In the “Introduction” section, the authors start to discuss about  tendon and ligament insertions, fibrocartilage and anterior cruciate ligament (ACL). Even if the paper is basically focused on the  contribution of Periostin to Fibrocartilage Layer Growth of Patella Tendon Tibial Insertion and the author report some works on the topic as well as interesting approach on tendons repair (Wang, Y.; Jin, S.; Luo, D.; He, D.; Shi, C.; Zhu, L.; Guan, B.; Li, Z.; Zhang, T.; Zhou, Y.; Wang, C.Y.; Liu, Y. Functional regeneration and repair of tendons using biomimetic scaffolds loaded with recombinant periostin. Nat Commun 2021, 12, 1293),  I also suggest to further stress and BRIEFLY report some interesting strategies towards the tendon repair/regeneration involving the role of 3D porous structures, cells and combination of materials properties and dynamic stimulations (e.g., Journal of Biomaterials Science, Polymer Edition, 2010, 21(8-9), pp. 1173–1190). Then,  the authors should continue to stress their approach on the study related to the Contribution of Periostin to Fibrocartilage Layer Growth of Patella Tendon Tibial Insertion in Mice. All of this should improve the quality of the paper, reporting important features as well as further methodologies  in designing and analyzing specific technical features, solutions to tendon problems and functional properties, thus helping the different kinds of readers to better understand the value of their work.

-        The Introduction and/or discussion section as well as the list of references should be improved according to the above reported comments. Some technical solutions concerning innovative solutions to tendon problems should be briefly reported.

-        The quality of some figures should be improved.

-        The title is adequate and appropriate for the content of the article.

-        The abstract contains information of the article.

-        Figures and captions are essential and clearly reported.

Author Response

Responses to Reviewer #1

Thank you for your comments. As you kindly suggested, we improved the manuscript. Colored words below are parts of the text we revised according to your suggestions.

Comment 1.

The paper deals with the Contribution of Periostin to Fibrocartilage Layer Growth of Patella Tendon Tibial Insertion in Mice, reporting interesting results. In the “Introduction” section, the authors start to discuss about tendon and ligament insertions, fibrocartilage and anterior cruciate ligament (ACL). Even if the paper is basically focused on the contribution of Periostin to Fibrocartilage Layer Growth of Patella Tendon Tibial Insertion and the author report some works on the topic as well as interesting approach on tendons repair (Wang, Y.; Jin, S.; Luo, D.; He, D.; Shi, C.; Zhu, L.; Guan, B.; Li, Z.; Zhang, T.; Zhou, Y.; Wang, C.Y.; Liu, Y. Functional regeneration and repair of tendons using biomimetic scaffolds loaded with recombinant periostin. Nat Commun 2021, 12, 1293), I also suggest to further stress and BRIEFLY report some interesting strategies towards the tendon repair/regeneration involving the role of 3D porous structures, cells and combination of materials properties and dynamic stimulations (e.g., Journal of Biomaterials Science, Polymer Edition, 2010, 21(8-9), pp. 1173–1190). Then, the authors should continue to stress their approach on the study related to the Contribution of Periostin to Fibrocartilage Layer Growth of Patella Tendon Tibial Insertion in Mice. All of this should improve the quality of the paper, reporting important features as well as further methodologies in designing and analyzing specific technical features, solutions to tendon problems and functional properties, thus helping the different kinds of readers to better understand the value of their work.

The Introduction and/or discussion section as well as the list of references should be improved according to the above reported comments. Some technical solutions concerning innovative solutions to tendon problems should be briefly reported.

Response 1.

Thank you for your comment. According to your comment, we added sentences about technical solutions of tendon-to-bone healing problems in Introduction and Discussion as follow.

Previous Manuscript: 1. Introduction, 4th paragraph

“Therefore, the purpose of this study is to clarify the influence of Periostin on the growth of fibrocartilage layers of the PT tibial insertion using Periostin knockout mice. We hypothesized that Periostin would influence the growth of fibrocartilage layers of PT tibial insertion.”

Revised Manuscript: 1. Introduction, 4th paragraph

“Therefore, the purpose of this study is to clarify the influence of Periostin on the growth of fibrocartilage layers of the PT tibial insertion using Periostin knockout mice. We hypothesized that Periostin would influence the growth of fibrocartilage layers of PT tibial insertion. The influence of Periostin on the growth of the patellar tendon insertion is expected to be important in the repair and regeneration of Periostin on the patellar tendon insertion.”

Previous Manuscript: 4. Discussion, 5th paragraph

“In terms of clinical diseases, Osgood-Schlatter disease is a well-known tendinopathy involving the PT tibial insertion [14], which requires prolonged treatment. Periostin has been reported to be up-regulated by growth factors such as BMP-2, basic FGF, TGF-β, and platelet-derived growth factor [19]. Periostin is upregulated during muscle regeneration [20] and fracture healing [21], and is necessary to regenerate tendons [22]. Periostin may be a candidate to treat tendinopathies such as Osgood-Schlatter disease. Moreover, Periostin may contribute as a marker of the repair/healing process and a predictor of treatment response.”

Revised Manuscript: 4. Discussion, 5th paragraph Line 692-705

“In terms of clinical diseases, Osgood-Schlatter disease is a well-known tendinopathy involving the PT tibial insertion [14], which requires prolonged treatment. Periostin has been reported to be up-regulated by growth factors such as BMP-2, basic FGF, TGF-β, and platelet-derived growth factor [22]. Periostin is up-regulated during muscle regeneration [23] and fracture healing [24], and is necessary to regenerate tendons [25]. The previously mentioned growth factors enhanced tendon-to-bone healing [26]. Moreover, dynamic ten-sile stimulation may be an essential factor for tendon regeneration [27]. In bone modeling and remodeling, mechanical stress plays an important role, and Periostin is expected to be involved in controlling these systems in bone [21]. Therefore, it is possible that Periostin influences growth factors and is involved in the growth, maintenance, and regeneration of tendons and bones in a mechanical environment. Periostin may be a candidate to treat tendinopathies such as Osgood-Schlatter disease. Moreover, Periostin may contribute as a marker of the repair/healing process and a predictor of treatment response, and ten-don-to-bone healing.”

We added references as follow.

  1. Kudo, A. Periostin in bone biology. Adv Exp Med Biol 2019, 1132, 43–47.
  2. Rodríguez-Merchán, E.C. Anterior cruciate ligament reconstruction: Is biological augmentation beneficial? Int J Mol Sci 2021, 22, 12566.
  3. Lee. J.; Guarino, V.; Gloria, A.; Ambrosio, L.; Tae, G.; Kim, Y.H.; Jung, Y.; Kim, S.H.; Kim, S.H. Regeneration of Achilles' tendon: the role of dynamic stimulation for enhanced cell proliferation and mechanical properties. J Biomater Sci Polym Ed 2010, 21, 1173–1190.

Comment 2.

The quality of some figures should be improved.

Response 2.

Thank you for your comment. According to your comment, we replaced Figure1D and E.

Comment 3.

The title is adequate and appropriate for the content of the article.

The abstract contains information of the article.

Figures and captions are essential and clearly reported.

Response 3.

Thank you very much.

Reviewer 2 Report

This is a companion paper to an in press report from the same authors (see Ortho Trauma Surg Res, ref 13). The in press paper examines the mostly the same parameters for the ACL while here those parameters are measured in the patella tendon insertion. Justification is provided as that the 2 tendons have different mechanical environments.

The study is straightforward and data are reasonably well presented. Conclusions are statistically based.

Problems:

1. Study methods: It is unclear from the histology presented what was measured. For instance the border between the patella and the tendon is not obvious.

2. Study methods: how were enthesis chondrocytes distinguished from tendon tenocytes?

3. Discussion: the mechanical environment and how it may differentially affect the patella tendon vs the ACL in normal and periostin KO mice is not discussed. The differences are noted, but there is no context as to why there are differences and more specifically whether periostin has a different or similar role between the two tendon types.

4. Discussion: why is there no Col2a1 staining prior to 4 weeks of age (Figure 2E) when chondrocytes (or are these cells tenocytes) are present, proliferating, secreting GAGs and during the time the junction between tendon and bone is forming?

5. Discussion: how does periostin regulate cell proliferation and apoptosis?

Line 23: Authors state that periostin decreased chondrocytes, when what the meant was that loss of Postn (KO mice) led to decreased chondrocytes.

Line 77: periostin KO genotype must have been confirmed by PCR analysis of isolated DNA and not by electrophoresis, as noted. (Indeed that methods section of this paper is very similar to the in press paper).

Figure 1D: appears green with only isolated cells showing some saf-O staining. Is this what is meant as saf-O stained area?

Figure 1E: the tidemark is not obvious.

Line 137: missing period.

Table 1: statistics should be re-checked, for instance under # of chondrocytes (tenocytes?)/width of insertion 10w data in which means are 256 vs 70 are not significant (and not normally distributed) while at 12 weeks means of 228 vs 141 are significant (p=0.014)?

Graphs: The data markers should be spread out to show all observations or the figure legends should explicitly state that all N=6 and that some values/markers may be overlapping.

Author Response

Responses to Reviewer #2

Thank you for your comments. As you kindly suggested, we improved the manuscript. Colored words below are parts of the text we revised according to your suggestions.

Comment 1.

  1. Study methods: It is unclear from the histology presented what was measured. For instance the border between the patella and the tendon is not obvious.

Response 1.

Thank you for your comment. According to your comment, we added sentences as follow.

Previous Manuscript: 2. Materials and Methods, 2.3. Histomorphometric Analysis

“Histomorphometric analysis was performed in accordance with the previous report [13]. Areas stained red by safranin O in the fibrocartilage layers of the PT tibial insertion were evaluated as GAG areas (Figure 1D) [13]. Brown stained areas of type II collagen were measured in the fibrocartilage layers [13]. The total length of tidemark stained with hematoxylin and eosin was measured (Figure 1E) [13].”

Revised Manuscript: Materials and Methods, 2.3. Histomorphometric Analysis

“Histomorphometric analysis was performed in accordance with our previous report [13]. Lower-density staining cartilaginous tissue with round cells between the hyaline car-tilage area and ligament were identified as the fibrocartilage layers at 1 d and 1 and 2 weeks of age [13]. In the other specimens, the fibrocartilage layers were identified as layers with round cells between the ligament and bone. [13]. The border between tendon and fi-brocartilage layers was identified by spindle-shaped cells and round cells [13]. In the fi-brocartilage layers of the PT tibial insertion, regions stained red by safranin O were identi-fied as GAG production areas (Figure 1D) [13]. Regions stained brown were identified as type II collagen [13]. The total tidemark length was measured using hematoxylin and eo-sin staining (Figure 1E) [13].”

Comment 2.

  1. Study methods: how were enthesis chondrocytes distinguished from tendon tenocytes?

Response 2.

Thank you for your comment. According to your comment, we added sentences as follow.

Previous Manuscript: 2. Materials and Methods, 2.3. Histomorphometric Analysis

“Histomorphometric analysis was performed in accordance with the previous report [13]. Areas stained red by safranin O in the fibrocartilage layers of the PT tibial insertion were evaluated as GAG areas (Figure 1D) [13]. Brown stained areas of type II collagen were measured in the fibrocartilage layers [13]. The total length of tidemark stained with hematoxylin and eosin was measured (Figure 1E) [13].”

Revised Manuscript: Materials and Methods, 2.3. Histomorphometric Analysis

“Histomorphometric analysis was performed in accordance with our previous report [13]. Lower-density staining cartilaginous tissue with round cells between the hyaline car-tilage area and ligament were identified as the fibrocartilage layers at 1 d and 1 and 2 weeks of age [13]. In the other specimens, the fibrocartilage layers were identified as layers with round cells between the ligament and bone. [13]. The border between tendon and fi-brocartilage layers was identified by spindle-shaped cells and round cells [13]. In the fi-brocartilage layers of the PT tibial insertion, regions stained red by safranin O were identi-fied as GAG production areas (Figure 1D) [13]. Regions stained brown were identified as type II collagen [13]. The total tidemark length was measured using hematoxylin and eo-sin staining (Figure 1E) [13].”

Comment 3.

  1. Discussion: the mechanical environment and how it may differentially affect the patella tendon vs the ACL in normal and periostin KO mice is not discussed. The differences are noted, but there is no context as to why there are differences and more specifically whether periostin has a different or similar role between the two tendon types.

Response 3.

Thank you for your comment. According to your comment, we added sentences about differentially affect the patella tendon vs the ACL as follow.

Previous Manuscript: 4. Discussion, 4th paragraph

“In comparison of a previous study that reported ACL insertion growth using Perios-tin KO mice [13], significant differences were observed in chondrocyte proliferation, chon-drocyte apoptosis, and the number of chondrocytes, even from 4 to 12 weeks in this study. Moreover, tidemark length of PT insertion showed a significant difference earlier than that of ACL insertion, and GAG production showed a significant difference at 8, 10 and 12 weeks in this study. Other phenomena were similar in both ACL and PT insertion. Both ACL and PT tibial insertion are direct-type insertions that include four transitional tissue layers: ligament or tendon, two fibrocartilage layers (unmineralized and mineralized), and bone that transmit mechanical stress [18]. However, in the ACL, both ends are made up of bone, whereas one end of the PT tibial insertion is a muscle through the patella. The differences between the ACL and PT tibial insertion can be due to the differences in struc-ture and mechanical environments. The PT tibial insertion is affected by muscle traction in the growth period and may subjected to greater tensile stresses than the ACL insertion.”

Revised Manuscript: 4. Discussion, 4th paragraph

“In comparison with a previous study that reported ACL insertion growth using Peri-ostin KO mice [13], significant differences were observed in chondrocyte proliferation, chondrocyte apoptosis, and the number of chondrocytes, even from 4 to 12 weeks in this study. Moreover, tidemark length of PT insertion showed a significant difference earlier than that of ACL insertion, and GAG production showed a significant difference at 8, 10, and 12 weeks in this study. Other phenomena were similar in both ACL and PT insertion. Both ACL and PT tibial insertion are direct-type insertions that include four transitional tissue layers: ligament or tendon, two fibrocartilage layers (uncalcified and calcified), and bone that transmit mechanical stress [18]. However, in the ACL, both ends are made up of bone, whereas one end of the PT tibial insertion is a muscle through the patella. The dif-ferences between the ACL and PT tibial insertion can be due to the differences in structure and mechanical environments. It has been reported that the actual 1/3 width of the bone-PT-bone complex has a higher ultimate load than the femur-ACL-tibia complex [19,20]. Therefore, the PT tibial insertion is affected by muscle traction in the growth period and may be subjected to greater tensile stresses than the ACL insertion. Moreover, Perios-tin is sensitive to mechanical stress, and controls bone modeling and remodeling [21]. Periostin is involved in the growth of the ACL insertion and the PT tibial insertion in a mechanical environment, however, Periostin may have a different influence on the ACL insertion than the PT tibial insertion where the mechanical load is larger.”

We added references as follow.

  1. Woo, S.L.; Hollis, J.M.; Adams, D.J.; Lyon, R.M.; Takai, S. Tensile properties of the human femur-anterior cruciate liga-ment-tibia complex. The effects of specimen age and orientation. Am J Sports Med 1991, 19, 217–225.
  2. Cooper, D.E.; Deng, X.H.; Burstein, A.L.; Warren, R.F. The strength of the central third patellar tendon graft. A biomechani-cal study. Am J Sports Med 1993, 21, 818–823.
  3. Kudo, A. Periostin in bone biology. Adv Exp Med Biol 2019, 1132, 43–47.

Comment 4.

  1. Discussion: why is there no Col2a1 staining prior to 4 weeks of age (Figure 2E) when chondrocytes (or are these cells tenocytes) are present, proliferating, secreting GAGs and during the time the junction between tendon and bone is forming?

Response 4.

Thank you for your comment. According to your comment, we added sentences about no Col 2 staining prior to 4 weeks of age when chondrocytes are present, proliferating, secreting GAGs and during the time the junction between tendon and bone is forming as follow.

Previous Manuscript: 4. Discussion, 2nd paragraph

“The GAG-stained area was high from day 1 to week 1 in WD, and from week 1 to 6 in KO. The GAG-stained area of KO was lower than that of WD at day 1 and week 1, 4, 8, 10 and 12. Absence of Periostin can delay and decrease GAG production. The type II collagen staining area in KO was lower than that in WD at 8 weeks. Absence of Periostin can also decrease type II collagen production. The percentage of tidemark length in KO was lower than that in WD at 4 and 6 weeks. Absence of Periostin can delay tidemark production. Periostin plays an important role in the growth of the ECM at the PT tibial insertion and prevents complete growth.”

Revised Manuscript: 4. Discussion, 2nd paragraph

“The GAG-stained area was high from day 1 to week 1 in WD, and from week 1 to 6 in KO. The GAG-stained area of KO was lower than that of WD at day 1 and week 1, 4, 8, 10 and 12. Absence of Periostin can delay and decrease GAG production. The type II collagen staining area in KO was lower than that in WD at 8 weeks. Absence of Periostin can also decrease type II collagen production. Although chondrocyte proliferation and increased GAG-stained areas were observed in the early stage of growth, it has been reported that type II collagen is observed after the calcified and uncalcified cartilage layers are formed at the insertion [12]. Therefore, type II collagen was expected to have been observed in the second half of the growth process. The percentage of tidemark length in KO was lower than that in WD at 4 and 6 weeks. Absence of Periostin can delay tidemark production. Periostin plays an important role in the growth of the ECM at the PT tibial insertion and prevents complete growth.”

Comment 5.

  1. Discussion: how does periostin regulate cell proliferation and apoptosis?

Response 5.

Thank you for your comment. According to your comment, we rephrased sentence as follow.

Previous Manuscript: 4. Discussion, 3rd paragraph

“Chondrocyte proliferation in KO was lower than that in WD at 1 day and 1, 4 and 8 weeks. Chondrocyte apoptosis in KO was also lower than that in WD 1 day and 4, 6, and 12 weeks. Although the number of chondrocytes gradually decreased in both groups, the number of chondrocytes in KO was also lower than that in WD at 1 day and 2, 8 and 12 weeks. Periostin can influence the activity of chondrocyte proliferation and apoptosis in PT tibial insertion. The imbalance between chondrocyte proliferation and apoptosis can affect the number of chondrocytes. As a result, it can lead to a delay in and incomplete ECM growth at the PT tibial insertion. In PT tibial insertion growth, Periostin is thought to be an important regulator.”

Revised Manuscript: 4. Discussion, 3rd paragraph

“Chondrocyte proliferation in KO was lower than that in WD at 1 day and 1, 4 and 8 weeks. Chondrocyte apoptosis in KO was also lower than that in WD 1 day and 4, 6, and 12 weeks. Although the number of chondrocytes gradually decreased in both groups, the number of chondrocytes in KO was also lower than that in WD at 1 day and 2, 8 and 12 weeks. Periostin may upregulate chondrocyte proliferation and apoptosis in PT tibial insertion. The imbalance between chondrocyte proliferation and apoptosis can affect the number of chondrocytes. As a result, it can lead to a delay in and incomplete ECM growth at the PT tibial insertion. In PT tibial insertion growth, Periostin is thought to be an im-portant regulator.”

Comment 6.

Line 23: Authors state that periostin decreased chondrocytes, when what the meant was that loss of Postn (KO mice) led to decreased chondrocytes.

Response 6.

Thank you for your comment. According to your comment, we rephrased sentence as follow.

Previous Manuscript: Abstract

“Conclusion: Periostin decreased chondrocyte proliferation, chondrocyte apoptosis, and the num-ber of chondrocytes in the growth process of PT tibial insertion. Moreover, Periostin decreased and delayed GAG and type II collagen production, and delayed tidemark formation in the growth process of PT tibial insertion. Periostin can, therefore, contribute to the growth of fibro-cartilage layers in PT tibial insertion. Periostin deficiency may result in incomplete growth of PT tibial insertion.”

Revised Manuscript: Abstract

“Loss of Periostin led to decreased chondrocyte proliferation, chondrocyte apoptosis, and the number of chondrocytes in the growth process of PT tibial insertion. Moreover, Periostin de-creased and delayed GAG and type II collagen production, and delayed tidemark formation in the growth process of PT tibial insertion. Periostin can, therefore, contribute to the growth of fi-brocartilage layers in PT tibial insertion. Periostin deficiency may result in incomplete growth of PT tibial insertion.”

Comment 7.

Line 77: periostin KO genotype must have been confirmed by PCR analysis of isolated DNA and not by electrophoresis, as noted. (Indeed that methods section of this paper is very similar to the in press paper).

Response 7.

Thank you for your comment. As you say, we made a mistake. We have corrected it as follow.

Previous Manuscript: 2. Materials and Methods, 2.1. Animal Preparation 2nd paragraph

“The Periostin knockout genetic confirmation with electrophoresis, using the tail DNA, was performed according to the previous report [13]. NucleoSpin® Tissue (Takara Bio Inc., Kusatsu, Shiga, Japan), and TaKaRa TaqTM Hot Start Version (Takara Bio Inc., Kusatsu, Shiga, Japan) were used. Forward primer was (F1):5’-TGAAGCTACCCATCTCCCAAATG-3’, and reverse primers were (R1):5’-CCTCTCCCAGCGTTCATAAATC-3’ and (R2):5’-ACCATCCTGTAGGCTCTTCAAAC-3 (Eurofins Genomics, Tokyo, Japan). Under the conditions described in the Mouse Conventional Knockout User Manual (Cyagen Bio-sciences Inc., Silicon Valley, CA, USA), the polymerase chain reaction was performed us-ing a thermal cycler (MiniAmp Plus, Thermo Fisher Scientific, Waltham, MA, USA) [13].”

Revised Manuscript: 2. Materials and Methods, 2.1. Animal Preparation 2nd paragraph

“The genetic confirmation of Periostin knockout by polymerase chain reaction (PCR) analysis of isolated DNA was performed using tail DNA according to our previous report [13]. NucleoSpin® Tissue (Takara Bio Inc., Kusatsu, Shiga, Japan), and TaKaRa TaqTM Hot Start Version (Takara Bio Inc., Kusatsu, Shiga, Japan) were used. Forward primer was (F1):5’-TGAAGCTACCCATCTCCCAAATG-3’, and reverse primers were (R1):5’-CCTCTCCCAGCGTTCATAAATC-3’ and (R2):5’-ACCATCCTGTAGGCTCTTCAAAC-3 (Eurofins Genomics, Tokyo, Japan). Under the conditions described in the Mouse Conventional Knockout User Manual (Cyagen Bio-sciences Inc., Silicon Valley, CA, USA), PCR was performed using a thermal cycler (MiniAmp Plus, Thermo Fisher Scientific, Waltham, MA, USA) [13].”

Comment 8.

Figure 1D: appears green with only isolated cells showing some saf-O staining. Is this what is meant as saf-O stained area?

Response 8.

Thank you for your comment. Safranin-O stained area is red stained area.

We replaced the Figure 1D.

Comment 9.

Figure 1E: the tidemark is not obvious.

Response 9.

Thank you for your comment. We replaced the Figure 1E.

Comment 10.

Line 137: missing period.

Response 10.

Thank you for your comment. But, there is a period in this sentence. Please confirm.

Previous Manuscript: Line 137

“If normality was not confirmed, the Mann-Whitney U test was used to compare data between WD and KO.”

Comment 11.

Table 1: statistics should be re-checked, for instance under # of chondrocytes (tenocytes?)/width of insertion 10w data in which means are 256 vs 70 are not significant (and not normally distributed) while at 12 weeks means of 228 vs 141 are significant (p=0.014)?

Response 11.

Thank you for your comment. According to your comment, we have re-checked the statistics. The part where there was a mistake in the data description has been corrected (In the number of chondrocytes per width of insertion at 12w in NO group). There were no mistakes in the description of the significant difference.

In the number of chondrocytes per width of insertion at 10w, there was no significant difference using the Mann-Whitney U test (P = 0.065).

In the number of chondrocytes per width of insertion at 12w, there was a significant difference using the Student t-test (P = 0.014). But, there was a mistake in the description of SD, so we corrected it. 140.7 ± 477.7âž¡140.7 ± 47.7

Comment 12.

Graphs: The data markers should be spread out to show all observations or the figure legends should explicitly state that all N=6 and that some values/markers may be overlapping.

Response 12.

Thank you for your comment. According to your comment, we added the sentence “Some markers may be overlapping.” in each figure legend. The descriptions of “n = 6” were already described.
